# Psychometric Properties of the Coparenting Relationship Scale in Ecuadorian Parents

**DOI:** 10.3390/ejihpe15070117

**Published:** 2025-06-25

**Authors:** Verónica Paredes, Andrés Ramírez, María José Rodríguez-Reyes, Luis Burgos-Benavides, Francisco Javier Herrero-Diez

**Affiliations:** 1Clinical Psychology Career, Catholic University of Cuenca, Cuenca 010107, Ecuador; vparedest@ucacue.edu.ec (V.P.);; 2Department of Clinical Psychology, Universidad Politécnica Salesiana, Cuenca 010107, Ecuador; aramirezc1@ups.edu.ec; 3Department of Psychology, University of Oviedo, 33003 Oviedo, Spain; herrero@uniovi.es

**Keywords:** psychometric, coparenting relationship scale, parenting, Ecuador

## Abstract

**Background:** The Coparenting Relationship Scale is a robust tool for assessing the quality of coparenting, with evidence of validity and reliability tested in diverse cultural and population contexts. **Objective:** The aim of this study was to evaluate the psychometric properties of the Coparenting Relationship Scale in Ecuadorian fathers and mothers in order to determine its validity and reliability in the specific cultural context of Ecuador. **Method:** An instrumental study was carried out by analyzing psychometric properties. The sample consisted of 867 participants from the province of Azuay, most of whom were women (66.8%), with a mean age of 35.4 years. Participants completed the Coparenting Relationship Scale. **Results:** The findings revealed adequate reliability of the scale in the Ecuadorian sample. In addition, factor analysis showed that the scale structure remained consistent in this sample, suggesting that the items grouped in the dimensions established by the original scale retain their validity in this specific context. Significant factor loadings consistent with the theoretical dimensions of the scale were observed, reinforcing its construct validity. The sixth model (M6), with three factors and 22 items, showed a good fit (*χ*^2^
*=* 399.43, *df* = 206, *χ*^2^/*df* = 1.94, *p* = 0.136), with adequate fit indices (CFI = 0.990, GFI = 0.996), a low approximation error (*RMSEA* = 0.033, 95% *CI* = [0.028–0.038]), and good residual fit (*SRMR* = 0.047). Additionally, the ECVI was 0.623. **Conclusions:** The Coparenting Relationship Scale is a valid and reliable instrument for assessing coparenting dynamics in Ecuadorian parents, providing a solid basis for interventions and policies aimed at strengthening family well-being in Ecuador.

## 1. Introduction

The family structure comprises various levels of interaction, among which the parental–filial relationship and the relationship between parents stand out. The latter plays a key role in the regulation of family interaction, since it encompasses the dynamics between parental figures, which can usually include marital or emotional issues (including the romantic and intimate relationship between adults) and joint parenting ([23]; [33]; [39]; [42]). An appropriate relationship between parents is the fundamental factor in adaptive family functioning, with a clear benefit for the children ([30]). The way in which parents or caregivers relate to each other in their parenting is called coparenting ([53]). Three decades of research have provided abundant scientific evidence to affirm that coparenting is one of the central relational processes of families ([64]).

Coparenting, conceptualized as a construct unlinked to gender role and family structure, refers to negotiated activities and relational aspects of caregivers working together to raise their children ([55]), coordinating parental tasks with each other ([39]). The main implication is that the partners share the responsibility of parenting and supporting the children, solving problems and making decisions together on different issues, with adequate communication and loyalty, while facilitating positive interactions between themselves and the children ([7]; [58]; [60]; [63]; [66]).

The quality of coparenting influences children during their learning and development process and tends to predict a more secure infant attachment, which affects the regulation of behaviors and emotions ([5]; [55]; [61]). It has also been shown that positive co-partnership is related to less problematic behaviors and better academic performance ([30]) It has also been found to be a key determinant of mental health and parental well-being. ([4]; [10]; [14]; [25]; [26]; [38]; [46]; [60]). On the contrary, inadequate parenting can cause emotional and behavioral problems in children ([57]). In particular, conflicting relationships between parents, the undermining of the figure of the other parent, the level of discord in coparenting, the triangulation of the children, subjecting them to parental interference or the disengagement of one of the parents in parenting ([20]; [56]; [63]; [66]) are associated with the appearance of internalizing and externalizing symptomatology in children. These dynamics increase stress and volatility in the family system, which negatively impacts the physical, mental, social and cognitive health of children, which can have long-term effects ([3]; [12]; [40]; [56]; [65]).

All this highlights the crucial role that the development and validation of measurement instruments has in practice, clinical, health evaluation and research, influencing decisions related to care, approach, and public policies ([15]). Therefore, the importance of assessing the psychometric properties and validity evidence of instruments and having culturally sensitive and validated measures available is an urgent need for certain contexts ([11]; [43]). At the international level, specific psychometric instruments have been developed to identify competencies associated with coparenting ([16]; [41]; [43]; [53]).

The Coparenting Relationship Scale (CRS), developed by [24] ([24]), has established itself as one of the most influential and highly regarded instruments within the scientific community ([57]). It is supported by strong empirical evidence demonstrating its reliability and validity ([11]), and has been translated and validated in multiple languages and countries. These include Chile ([16]), China ([30]), Spain ([50]; [57]), France ([21]), Indonesia ([1]), Portugal ([34], [35]; [49]), Romania ([19]), Sweden ([37]), as well as Belgium, Brazil, Israel, Italy, Japan, Switzerland, Turkey, and the United States ([64]). The CRS has proven effective across a wide range of populations, including divorced parents and their children ([34]), men in the prenatal period ([49]), fathers with children from single or multiple relationships ([37]), parents of preschool-aged children ([59]), women ([35]), heterosexual individuals and those with diverse gender identities ([17]; [36]), non-Hispanic white families ([31]; [67]), and parents with non-standard employment ([26]). Additionally, efforts are underway to adapt the scale to assess coparenting from the perspective of sons and daughters ([8]).

[24] ([24]) took as a frame of reference the ecological model of the [22] ([22], [23]) for the design and validation of the CRS. This results in a comprehensive self-report measure of parenting quality, which allows for the exploration of nuanced associations between parenting and child outcomes ([19]). The ecological model for understanding the coparenting relationship proposes four domains ([24]). The first domain (*childrearing agreement*) does not refer to the dynamics between parents, but to their views on how to raise a child, i.e., whether they are similar or not. The second domain (*division of labor*) focuses on how parents share and coordinate parenting tasks and their feelings about whether or not that division is equitable. The third domain (*joint management of family dynamics*) refers to how parents establish norms and expectations of behavior in the family and how they manage them. The fourth domain (*coparental support/undermining; coparental support*) includes one parent’s recognition of the other parent’s parental authority, parenting skills, and respect for the other parent’s contributions and decisions within the family; conversely, the weakening of coparenting (*coparental undermining*) is produced by undermining the other parent through criticism, contempt and blame. In addition, [24] ([24]) in the construction of the CRS included the nurture-based closeness dimension (*parenting-based closeness*), that “is conceptually related but distinct from coparental support. Whereas coparental support relates to respecting and upholding the other parent’s decisions, parenting-based closeness relates to the shared celebration of the child’s attainment of developmental milestones, the experience of working together as a team, and witnessing one’s partner develop as a parent” ([24]). These domains, which make up the CRS, are completed with the domain of *support/undermining* which is a product of three subscales: coparenting support, endorsement partner parenting and coparenting undermining.

The resulting CSR questionnaire is used as a measure of coparenting in several countries. This is evidenced by recent research from [64] ([64]), which included data from studies conducted in 10 countries and confirms an unchanging six-factor structure: *coparenting agreement, coparenting closeness, exposure to conflict, coparenting support, coparenting undermining and endorsed partner parenting, leaving out the division of labor*. The results conclude that the invariance of the dimensions of shared parenting and the determinants of its universality should be further investigated. These findings, in turn, are in line with those of [16] ([16]), who obtained a three-factor model based on the Chilean population, which also did not confirm the *division of labor*.

Therefore, the main objective was to evaluate the psychometric properties of the Coparenting Relationship Scale (CRS) in Ecuadorian parents, with the purpose of establishing its reliability, validity, and invariance as a measurement instrument within the specific cultural context of Ecuador. It was hypothesized that the Coparenting Relationship Scale (CRS) would demonstrate adequate reliability and validity when applied to Ecuadorian parents. Specifically, it was expected that the instrument would show high internal consistency and acceptable construct validity based on factor analysis. Additionally, measurement invariance was hypothesized across key demographic variables—namely, sex (mothers vs. fathers), type of couple relationship (married/cohabiting vs. separated/divorced), and parental rupture status.

## 2. Materials and Methods

### 2.1. Design

The instrumental research was carried out in two phases, in accordance with current standards for the validation of educational and psychological tests ([2]), as well as the established guidelines for the adaptation and translation of existing instruments ([6]; [18]; [45]).

### 2.2. Participants

The 867 study participants, selected through a non-probabilistic convenience sampling method ([62]), had a mean age of 35.39 years (SD = 8.401) and a median age of 35, indicating that half of the participants were younger than 35 and the other half were older. The age distribution showed a slight positive skewness, suggesting a higher concentration of ages below the mean: 25% of the participants were under 30 years of age, 50% were under 35, and 75% were under 40. Regarding sex, 66.8% of the participants were women and 33.2% were men. Concerning marital status, 62.1% were married, 18.6% were in a common-law partnership, 12.1% were divorced, 4.7% were single, and 2.5% were widowed. Overall, 80.6% reported having a partner, indicating that the majority of participants were in a stable relationship, a condition that could have influenced some of the variables analyzed in the study. When comparing age based on whether participants had a partner or not, it was observed that those with a partner had a higher mean age (mean = 35.83; median = 35) than those without a partner (mean = 33.59; median = 31). Additionally, the age dispersion was wider in the group without a partner (standard deviation = 9.02; interquartile range = 13) compared to those with a partner (standard deviation = 8.19; interquartile range = 10), indicating greater variability in the ages of people without a partner. The age distribution in both groups showed a slight positive skewness, which was somewhat greater in the group without a partner (skewness = 0.655) compared to the group with a partner (skewness = 0.565). Moreover, in both cases, the Shapiro–Wilk normality test was significant (*p* < 0.001), indicating that the age variable does not follow a normal distribution in either group.

### 2.3. Procedure

In the first instance, a convenience sampling was performed ([62]). The inclusion criteria encompassed parents residing in Ecuador who were involved in a coparenting relationship and who have at least one minor child. Parents who were not participating in a coparenting relationship or those with situations that could bias the results (e.g., significant legal conflicts) were excluded.

Nine educational institutions located in Cuenca (Ecuador) participated. Potential participants were contacted to participate in the research and those who gave their informed consent were invited, in small groups, to a classroom in the institutions where they received a booklet containing the ad hoc survey and the Coparenting Relationship Scale. The scale was applied in groups of no more than 15 participants. The researchers were available to answer any questions. Once the answer booklet was completed, a space was provided for participants to submit their answers, thus guaranteeing anonymity.

Ethical considerations sought to guarantee integrity and respect for the participants. The confidentiality of information was a priority, and an ethical protocol was established to manage possible situations of emotional distress or emerging conflicts during the study. Counseling and support resources were offered to participants who might feel affected by the topic of coparenting.

The objectives, procedures, and possible associated risks and benefits were clearly explained to the participants, and measures to protect privacy were implemented. Codes or identifiers were used to disassociate the data from personal information, in addition to ensuring the voluntariness of participation by allowing parents to withdraw at any time without negative consequences.

The present study followed the highest possible ethical standards, in accordance with the Helsinki declaration and institutional research ethics. Formal ethical approval was granted by the Human Research Ethics Committee of the Catholic University of Cuenca. Formal written informed consent was obtained from each participant prior to data collection. The informed consent form clearly documented the goals, objectives, procedures, risks and benefits, and the option to participate in the study.

### 2.4. Measures

The measures used were the *sociodemographic ad hoc questionnaire,* in which we requested information on several variables: sex, age of parent, marital status, and relationship status, and the *Coparenting Relationship Scale* ([24]), whose translation and transculturation of the instrument starts with the translation of [57] ([57]) and verified the vocabulary and forms of expression from one culture to another. The scale consists of 35 items, divided into 7 coparenting subscales: (1) coparenting agreement (4 items), (2) coparenting support (6 items), (3) endorse partner parenting (7 items); (4) coparenting undermining (6 items), (5) exposure to conflict (5 items), (6) division of labor (2 items), and (7) coparenting closeness (5 items), that serve to more accurately quantify coparental dynamics in family settings. It uses a 6-point Likert response scale (from 0 “not true in our case” to 6 “very true in our case”, except for the exposure to conflict subscale, where responses range from 0 “never” to 6 “very often”). For the correction, the mean score of the items of each of the subscales is calculated (there are inverse items). Example items are as follows: “I think my partner is a good parent”; “My partner and I have the same goals for our child”; “My partner does not trust my parenting skills”. Its application has been extended to several countries, offering good internal consistency. The Cronbach’s alpha reported in the study by [24] ([24]) ranges from 0.91 to 0.94. In contrast, the research by [57] ([57]) found Cronbach’s alpha values ranging from 0.83 to 0.94, while McDonald’s ordinal omega ranged from 0.83 to 0.84. Additionally, the reported fit indices were χ^2^/df = 1.04; CFI and TLI equal to 1; SRMR = 0.034; and RMSEA = 0.006.

### 2.5. Data Analysis

We conducted an instrumental study, following current standards for the validation of educational and psychological evaluations ([2]). The purpose of the statistical analysis is to evaluate the psychometric properties, validity, and reliability of the *Coparenting Relationship Scale (CSR)*, using advanced techniques with the software *R version 4.4.3, JASP version 0.19.3*, and Jamovi version 2.6.26.

Descriptive analyses are performed to examine the distribution of responses and to calculate means and standard deviations. This approach allowed us to obtain an overview of the perception of coparenting in the sample. Prior to factor treatment, we conducted data-to-data screening, in which the distribution of the data is analyzed, as well as the assumptions for factor treatment. The multivariate normality of the data was analyzed using the Mardian test. To verify these assumptions, we analyzed the residuals resulting from subjecting our data to a linear regression with a set of random numbers. If the distribution of the residuals from a spurious regression showed anomalies, this could be due to our data ([32]). To explore the underlying structure of the scale, an exploratory factor analysis was performed. This technique, with the help of specialized software, made it possible to identify possible latent dimensions or factors that contribute to the measurement of coparenting, offering a deeper understanding of the internal structure of the scale.

To analyze the structural validity of the scales, we performed a factor analysis using the R lavaan package ([54]). Discriminant validity was assessed by calculating the Heterotrait–Monotrait Ratio (HTMT) based on the mean of the item-to-factor correlations, using a criterion below 0.90, to indicate discriminant validity ([29]; [44]; [51]). Convergent validity was assessed by calculating the average variance extracted (AVE) and its square root to estimate the common variance between items and factors. According to [44] ([44]), AVE values equal to or greater than 0.37 were considered acceptable, although some authors recommend values of 0.50 or higher ([29]). Due to the absence of multivariate normality and the ordinal nature of our data, we used the diagonally weighted least squares method ([28]). The reliability of the data was analyzed using Cronbach’s *α* and McDonald’s *ω* ([52]). The incremental criteria used for the invariance were *CFI <* 0.010 and RMSEA < 0.015 ([13]). Finally, a correlation analysis was performed using Spearman’s coefficient to determine the associations between the different factors and exposure to conflict, since the distribution was not normal.

## 3. Results

The analysis of the distribution of responses within the analyzed sample allows a descriptive analysis of the CRS items. In general, the items showed high scores, with medians between 0 and 5 and means between 1.033 and 4.881. Several items exhibited negative skewness, indicating a greater concentration of responses at the high end of the scale. Similarly, kurtosis values reflected varying degrees of skewness in the distribution of the data. In terms of dispersion, standard deviations ranged from 1.555 to 2.480, demonstrating variability in participants’ responses.

The data did not reflect normality (*p* < 0.001 in none of the cases), and the analysis of items 1 to 30 revealed a consistent pattern of high scores across the sample; this suggests a generally positive perception of coparenting among the participants. The presence of negative skewness in several items indicated a tendency for responses to cluster at the high end of the scale. In addition, kurtosis values showed varying degrees of concentration in the data, with some items exhibiting a more pronounced peak; this suggests to us that most participants reported favorable experiences related to coparenting dynamics (for more information, see Appendix A). In addition, items 31, 32, 33, 33, 34, and 35 were recommended for exclusion due to the small sample size for these specific items. Since the sample did not show any structural breaks in the data, it was advised to eliminate these items to ensure the overall validity of the analysis.

Table 1 presents the different dimensionality models of the Coparenting Relationship Scale in Ecuadorian parents, evaluating the statistical adjustments after the elimination of problematic items. The factor structure of Model 1 presented a poor fit. In model 2, items 5 and 20 were eliminated due to their low factor loadings; the fit indices improved, but were still unsatisfactory. In model 3, item 6 was eliminated because it presented a significant loading among several factors, according to the modification values (819.850 in Factor 6, 789.416 in Factor 4 and 775.373 in Factor 2), the parameters improved considerably. In Model 4, items 7, 28 and 29 were eliminated due to their low factor loadings, this produced a significant improvement in the model fit, the fit indices were good, with a CFI of 0.992 and an RMSEA of 0.031 (CI: 0.027–0.036).

The results showed a high intercorrelation between Factors 2, 4 and 6 (values above 0.9), indicating the measurement of similar constructs. Therefore, we moved to Model 5, where the three intercorrelated factors are one, keeping Factor 1 and Factor 5 unchanged, the fit indices continued to be robust. In Model 6, items 18 and 23 were eliminated, the final fit was good and confirmed that the simplified structure was adequate and represented in a manner consistent with the data. Factor loadings for the coparenting agreement factor ranged from 0.734 to 0.763, those for the coparenting closeness, support and endorse partner factor ranged from 0.624 to 0.893, and those for the coparenting undermining factor ranged from 0.529 to 0.832.

Table 2 shows the measures of invariance according to sex, partner, and rupture. For these three aspects, the model presents total levels of invariance (configural, metric, scalar, and strict), which shows that the measurement is consistent among the different groups with strong implications at a practical–professional level.

Table 3 presents the loadings, reliability and validity (convergent and discriminant) of the three-factor model of the Coparenting Relationship Scale in Ecuadorian parents. In general, the factor loadings, reliability and discriminant validity of the three factors were adequate. Convergent validity was adequate for the coparenting agreement and coparenting closeness and support and endorse partner factors, explaining more than 50% of the variance. The coparenting undermining factor presented adequate convergent validity, explaining less than half of the variance, but more than 37%.

Table 4 presents a descriptive analysis of the sums of the factors of the three-factor model and the exposure to conflict variable. On the other hand, a low positive correlation was observed between Factor 1 (coparenting agreement) and Factor 2 (coparenting closeness, support and endorse partner) (*ρ* = 0.122, *p* < 0.001), suggesting that coparenting agreement is associated with an increase in coparenting closeness, support and endorse partner, although the relationship is slight. There was also a negative correlation between Factor 1 (coparenting agreement) and Factor 3 (coparenting undermining) (*ρ* = −0.356, *p* < 0.001), indicating that a higher level of coparenting agreement reduces coparenting undermining. Similarly, the relationship between Factor 2 and Factor 3 was negative and moderate (*ρ* = −0.399, *p* < 0.001), suggesting that greater mutual support is associated with less fragility in the shared parenting relationship.

In terms of sex, Table 5 presents no statistically significant differences were found between males and females. In both groups, most participants showed robust coparenting, with a higher proportion in females compared to males. The presence of a partner was significantly associated with coparenting, with a considerable effect size. Those with a partner showed mostly robust coparenting, whereas those without a partner had a more even distribution across coparenting levels, with similar proportions in the inadequate and moderate categories. These results suggest that coparenting was favored by the presence of a partner, while sex did not play a determining role in its distribution.

## 4. Discussion

The present study offers the results of the adjustment process of the dimensionality of the Coparenting Relationship Scale in Ecuadorian parents, evaluated through several models. Initially, Model 1, which included six factors and 30 items, was tested, with poor statistical fit indices, with a χ^2^ high and a low ratio of χ^2^/gl, accompanied by a CFI and an RMSEA which indicated that the original structure ([24]) did not adequately represent the data. This result suggested the need for additional adjustments to improve the factor structure ([67]). This led to the creation of Model 2, where the number of items was reduced to 28 by eliminating 2 items with low reliability; although the fit indices improved slightly, they were still unsatisfactory, which prompted further refinement of the model. The next step involved eliminating an additional item (Model 3) because it presented significant loadings in several factors and generated problems in the interpretation of the structure, after the elimination, Model 3 presented a considerable improvement in the adjustment indexes, approaching an acceptable level ([9]). To further optimize the scale, Model 4 eliminated three additional items that had low factor loadings, resulting in a significant improvement in model fit. Finally, in Model 5, three highly correlated factors were unified, which allowed the structure to be simplified to three factors with 24 items, maintaining an excellent statistical fit. In the last step, Model 6 eliminated 2 more items, leaving a final structure of three factors and 22 items that presented an optimal fit and a coherent representation of the data ([31]).

Secondly, the invariance analysis to evaluate the consistency of the model between different groups showed that the factor structure was invariant at all levels of comparison, both by sex, partner status and breakup experience. This indicated that the organization of the items and factors was similar between men and women, as well as between people with and without a partner, and between those who had and had not experienced a breakup ([20]). The fit indices of the configural, metric, scalar, and strict model confirmed the equivalence of factor loadings, intercepts and measurement errors between the different groups.

In terms of reliability and validity of the final three-factor model, it was found that Factor 1, related to coparenting agreement, showed adequate factor loadings and good internal consistency, evidenced by acceptable Cronbach’s alpha and omega coefficients. This factor also showed adequate convergent and discriminant validity, indicating that it was distinct from the other factors.

Finally, results on coparenting levels indicated that the majority of participants exhibited robust coparenting, while a small proportion showed moderate or inadequate levels of collaborative parenting. These findings suggested that while most parents shared responsibilities effectively, there were some areas that could benefit from interventions to improve the coparenting closeness and support and endorse partner dimensions. This is crucial, as interventions focused on improving coparenting closeness and reducing coparenting undermining can be more effective and direct, facilitating positive change in coparenting relationships ([48]).

Comparing these results with the studies of [24] ([24]), there was a tendency to identify a small number of factors that adequately reflect the complexity of coparenting. For example, the study of [30] ([30]) in the Chinese context found Cronbach’s alpha coefficients ranging from 0.69 to 0.93 for the six subscales, indicating that all subscales were adequate. These results are comparable to those obtained in the present investigation, which demonstrated satisfactory reliability in the dimensions analyzed.

In addition, other studies, such as that of [31] ([31]), in the context of white, non-Hispanic families, and that of [9] ([9]), in the Italian context, also corroborate the reliability of the measure, finding Cronbach’s alphas ranging between 0.60 and 0.88 for mothers and between 0.71 and 0.85 for fathers. The consistency in results across diverse populations reinforces the idea that CSR is an effective tool for measuring coparenting in different cultural contexts.

The favorable fit indices observed in Models 3 and 4 not only reinforce the construct validity of the Coparenting Relationship Scale but also suggest that a more simplified structure may be more accessible and useful for professionals working with families. The ability to efficiently measure coparental relationships can facilitate the development of more effective intervention strategies ([20]), focusing on the most relevant factors.

Regarding the convergent validity of Factor 3, the results show that its AVE (0.490) is slightly below the recommended threshold of 0.5, suggesting that the factor does not optimally explain the variance of its items. This finding may be related to the low factor loading of item 16 (0.529), highlighting the need for a more detailed analysis of the relevance of this item within the assessed construct. For future research, it is recommended to consider reformulating or eliminating this item, as well as exploring possible modifications in the factor structure to improve convergent validity.

Regarding the coparenting undermining (CU) dimension, its AVE value was 0.490. Although this value falls below the traditional threshold of 0.50 proposed by [27] ([27]), it is deemed acceptable according to the more flexible criterion proposed by [44] ([44]), who suggests that values equal to or greater than 0.37 may be adequate in applied research or in contexts where the model presents other strong indicators of validity and reliability. In this case, the AVE of CU, together with its high reliability indices (α = 0.853; ω = 0.836; CR = 0.852), supports the convergent validity of the dimension, indicating that a considerable proportion of the item variance is explained by the latent construct. This suggests that, despite not meeting the stricter criterion, the value is sufficient to justify the inclusion of the dimension in the model, especially in contexts such as Ecuador, where cultural adaptation of scales may influence the explained variance.

On the other hand, the identification of two factors in Model 4 highlights the importance of the dimensions of closeness and impairment, which could guide therapists and family counselors in designing programs that seek to strengthen relationships between parents. Interventions that directly address these dimensions can have a significant impact on family well-being, fostering more effective and collaborative shared parenting ([47]; [48]).

For future research, the study’s limitation that the sample is restricted to the province of Azuay (Ecuador) should be addressed, which could affect the generalizability of the results and would not reflect cultural and socioeconomic diversity, which could influence the applied practical orientation derived from the scale. It is also recommended to use the structural invariance model for the Coparenting Relationship Scale, which comprises three factors: coparenting agreement, coparenting closeness, support and endorse partner, and coparenting undermining. This approach will allow us to analyze whether the factor structure of the scale remains stable across different study groups, which will facilitate the validation of its applicability in diverse contexts and populations. Furthermore, it will improve the accuracy of assessing coparenting dynamics, which is especially relevant in countries such as Ecuador. In this country, there are alarmingly high divorce rates, currently ranking second nationwide, along with an increasing number of femicide cases and growing concerns about dysfunctional coparenting relationships. These social problems underscore the urgent need for reliable, valid, and culturally sensitive assessment tools that can support early detection, prevention strategies, and the design of targeted psychological and family interventions.

## 5. Conclusions

The study represents a significant and timely effort to validate the Coparenting Relationship Scale within the Ecuadorian context, addressing an urgent need for culturally relevant assessment tools in family psychology. The findings offer promising implications for professional practice, not only by enhancing the accuracy and sensitivity of evaluations in clinical and mediation settings, but also by contributing to the evidence base necessary for developing context-specific interventions. The validated scale can guide the design of targeted programs within clinical psychology, social work, and family mediation frameworks, helping professionals detect risk factors, improve communication between caregivers, and support the development of healthier family environments.

Moreover, the results of this study can inform and strengthen public policies aimed at supporting families and children in vulnerable situations. This includes the elaboration of evidence-based guidelines for family support services, the training of professionals working in family systems, and the promotion of collaborative parenting practices that contribute to the emotional, psychological, and social well-being of all family members. In a country like Ecuador, where divorce rates are high, coparenting conflicts are common, and gender-based violence including femicide remains a pressing concern, the availability of valid and reliable tools is not only important—it is essential.

From a psychometric perspective, the scale demonstrates excellent reliability indices, as well as factorial invariance across key sociodemographic groups, supporting its robustness and adaptability for diverse populations. These strong psychometric properties indicate that the scale can be used with confidence in both research and applied settings, and it holds potential for future longitudinal studies, cross-cultural comparisons, and intervention evaluations.

In conclusion, the findings, together with existing evidence in the literature, underscore the importance of the Coparenting Relationship Scale as a valuable tool for assessing and understanding coparenting dynamics. The differences obtained in the factorial structure reinforce the need, even between countries sharing the same language, for psychological instruments to be subjected to a process of adaptation and validation to the contexts in which they are to be used ([11]; [43]). As research in this area continues, it is essential to consider cultural diversity and the nuances of family dynamics in the development of future assessment tools and intervention programs. Therefore, adapting and validating this measure in different cultural and demographic contexts is critical, and identifying simplified models that adequately reflect these relationships can improve interventions and support for families.

## Figures and Tables

**Table 1 ejihpe-15-00117-t001:** Selection of the Coparenting Relationship Scale dimensionality models in Ecuadorian parents.

	Factors	Items	*x* ^2^	*df*	*x*^2^/*df*	*p*	*CFI*	*GFI*	*RMSEA*	*RMSEA [IC 95%]*	*SRMR*	*ECVI*
M^1^	6	30	4756.78	390	12.2	<0.001	0.857	0.968	0.114	[0.111–0.117]	0.113	5.755
M^2^	5	28	3675.77	340	10.81	<0.001	0.885	0.974	0.107	[0.104–0.110]	0.104	4.477
M^3^	5	27	2515.38	314	8.01	<0.001	0.918	0.981	0.090	[0.087–0.093]	0.088	3.126
M^4^	5	24	444.26	242	1.84	<0.001	0.992	0.996	0.031	[0.027–0.036]	0.045	0.705
M^5^	3	24	482.54	249	1.94	<0.001	0.990	0.996	0.033	[0.029–0.037]	0.047	0.733
M^6^	3 *	22 *	399.43	206	1.94	0.136	0.990	0.996	0.033	[0.028–0.038]	0.047	0.623

Note. * The model with the best fit, expected cross validation index (ECVI) was the lowest value. For more details on the results of Model 6 (M^6^), see the following link: https://n9.cl/4vvz0 (accessed on 12 January 2025).

**Table 2 ejihpe-15-00117-t002:** Invariance of Model 6 of the Coparenting Relationship Scale in Ecuadorian parents.

	Model	CFI	ΔCFI	RMSEA *[IC 95%]*	ΔRMSEA	SRMR	ΔSRMR
Sex							
	M_C_	0.995		0.023 [0.015–0.030]		0.053	
	M_C_ → M_M_	0.990	0.005	0.033 [0.027–0.038]	0.010	0.059	0.006
	M_M_ → M_S_	0.989	0.001	0.034 [0.028–0.039]	0.001	0.061	0.002
	M_S_ → M_E_	0.989	0.000	0.033 [0.027–0.038]	0.001	0.062	0.001
Couple							
	M_C_	0.994		0.024 [0.016–0.030]		0.048	
	M_C_ → M_M_	0.991	0.003	0.028 [0.022–0.034]	0.004	0.051	0.003
	M_M_ → M_S_	0.987	0.004	0.033 [0.028–0.038]	0.005	0.053	0.002
	M_S_ → M_E_	0.980	0.007	0.041 [0.036–0.045]	0.008	0.061	0.008
Rupture							
	M_C_	0.995		0.022 [0.013–0.028]		0.047	
	M_C_ → M_M_	0.991	0.004	0.029 [0.023–0.035]	0.007	0.051	0.004
	M_M_ → M_S_	0.989	0.002	0.031 [0.026–0.037]	0.002	0.053	0.002
	M_S_ → M_E_	0.985	0.004	0.036 [0.031–0.041]	0.005	0.058	0.005

*Note.* M_C_: metric invariance; M_M_: configural invariance; M_S_: scalar invariance; M_E_: strict invariance.

**Table 3 ejihpe-15-00117-t003:** Factor loadings, reliability, and convergent and discriminant validity of Model 6 of the Coparenting Relationship Scale in Ecuadorian parents.

	Reliability			AVE	HTMT		
	α	ω	*CR*		CA	CC-EP	CU
CA	0.794	0.794	0.790	0.563			
CC-EP	0.995	0.995	0.956	0.626	0.092		
CU	0.853	0.836	0.852	0.490	0.415	0.245	0.490

*Note.* CA: coparenting agreement; CC-EP: coparenting closeness, support and endorse partner; CU: coparenting undermining. The AVE was acceptable (CU) according to the ≥0.37 criterion proposed by [44] ([44]).

**Table 4 ejihpe-15-00117-t004:** Description of the sum of the factors of Model 6.

	CA	CC-EP	CU
Median	9	64	7
Mean	9.39	57.28	9.53
Standard deviation	5.25	19.76	9
Skewness	−0.07	−1.27	0.93
Standard error of Skewness	0.083	0.08	0.08
Kurtosis	−1.04	0.8	−0.04
Standard error of Kurtosis	0.17	0.17	0.17
Shapiro–Wilk	0.96	0.85	0.89
*p*-value de Shapiro–Wilk	<0.001	<0.001	<0.001
Minimum	0	0	0
Maximum	18	78	36
Percentile-25	5	48	2
Percentile-50	9	64	7
Percentile-75	14	72	15
*n*	867	867	867

*Note.* CA: coparenting agreement; CC-EP: coparenting closeness, support and endorse partner; CU: coparenting undermining.

**Table 5 ejihpe-15-00117-t005:** Coparenting (inadequate, moderate and robust) as a function of sex and partner.

Category	Inadequate	Moderate	Robust	*x* ^2^	*df*	*p*	*Cramér’s V*
Female	51 (5.88%)	41 (4.73%)	487 (56.17%)	3.588	2	0.166	0.064
Male	18 (2.08%)	14 (1.61%)	256 (29.53%)				
Without Partner	49 (5.65%)	50 (5.77%)	69 (7.96%)	346.334	2	<0.001	0.632
With Partner	20 (5.77%)	5 (1.44%)	674 (77.74%)				

## Data Availability

The database can be obtained by writing an e-mail to the first author of the article. The data in this study are accessible to anyone who needs to consult them.

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
