# Peer review of "Psychometric Properties of the Coparenting Relationship Scale in Ecuadorian Parents"

_ejihpe, 2025, doi:10.3390/ejihpe15070117_

Round 1
Reviewer 1 Report
Comments and Suggestions for Authors
REVIEW REPORT
Abstract: Overall, the abstract is well-structured, but with some minor adjustments, it could further enhance its effectiveness and precision. It follows the IMRD format and includes background, objective, method, results, and conclusions, which facilitates understanding of the study. However, although it mentions that the scale has been validated in various cultural contexts, it would be useful to include a brief comparison with previous studies in other contexts to highlight the novelty of the study. The abstract also mentions that the factor analysis showed consistency, but it could benefit from a brief explanation of what type of factor analysis was conducted and how the factor loadings were interpreted. Although the composition of the sample is mentioned, information on how participants were selected and whether there were any inclusion or exclusion criteria could be added. It would also be useful to include a brief mention of the practical implications of the findings for professionals working in the field of coparenting in Ecuador and the family welfare policies that may derive from it.
Keywords: The keywords are quite appropriate and reflect the main themes of the study well. However, with some adjustments, they could improve the visibility and precision of the article. For example, adding keywords that reflect the specific cultural context of the study, such as "Ecuador" or "Ecuadorian cultural context," could be beneficial. The authors might also consider including keywords that reflect the theoretical dimensions of the scale, such as "Closeness of coparenting," "Support of coparenting," and "Undermining of coparenting." Including a keyword related to the methodology used, such as "factor analysis" or "psychometric properties," could also be beneficial to attract researchers interested in these aspects.
Introduction: The introduction provides a broad and well-founded context on the importance of coparenting and its impact on family dynamics, highlighting the need for specific tools to assess this dynamic in different cultural contexts. This justifies the interest in the research. Numerous studies and recognized authors in the field are cited, which reinforces the credibility and academic rigor of the work. The introduction is solid and well-founded; however, with some adjustments in writing and structure, it could be even more effective. I would recommend a section more specifically focused on the current study, highlighting its objective and unique contribution to the field. Some citations appear to be from very recent studies, which is excellent, but the authors should ensure that all sources are relevant and up-to-date until 2025, when the article will be published.
There is a lack of specific information on the research questions and/or specific objectives of the study.
Method: The research design is confused with the sample selection system. The method employed should be stated and justified according to specific literature.
Participants: Although the sample is large, it is limited by the choice of convenience sampling. There is a lack of information demonstrating that it is of sufficient size, and the margin of error/confidence level or statistical power assumed for making possible inferences to the reference population. A sampling, at least by quotas of types of family systems, would have improved it.
Measurement Instrument: The description of both instruments could be improved, especially regarding the coparenting relationship scale. It should be completed with information on the psychometric properties of the original scale, as well as its use in other samples from previous studies with Spanish-speaking populations or other cultures where it has been used. The theoretical justification for its choice is also insufficiently addressed in the introduction.
Data Collection Procedure: Although consent was obtained, anonymity and confidentiality of data were ensured, and the necessary ethical principles for research with human beings were followed, the description of this aspect could be improved to avoid redundancies and make the description of how it was developed more precise.
Data Analysis and Results: In general, the data analysis is thorough and well-founded, but with some adjustments, it could be improved. A brief explanation of each advanced technique used could help readers less familiar with these methods to better understand the results. I also suggest including specific results of the Mardian test and how these influenced the data treatment to provide a more complete view. Additionally, I recommend making the confirmatory data analysis more explicit and including more details about the specific fit indices and how these supported the model structure. The convergent validity of Factor 3 (Coparenting Undermining) is slightly below the recommended threshold. It is suggested to review the items of this factor to improve its validity. All of this would significantly strengthen the validity of the study and provide a solid basis for future research. Including additional graphs or tables to visualize the results more clearly and accessibly could improve the presentation of the data.
Discussion: The discussion is well-structured and follows a logical sequence that facilitates understanding of the results and their interpretation. A detailed comparison with previous studies is made, which reinforces the validity of the findings and places the results in a broader context. It acknowledges the restriction of the sample to the province of Azuay as a study limitation, demonstrating a critical reflection on the results. The practical implications of the findings are highlighted, suggesting specific interventions to improve coparenting, which is useful for professionals in the field. Clear recommendations for future research are provided, which can guide subsequent studies and improve the understanding of coparenting in different contexts. While all of this is true, there are also aspects that could be improved within this section: The discussion mentions that the convergent validity of Factor 3 is slightly below the recommended threshold. It would be useful to delve deeper into the possible reasons for this deficiency and suggest specific strategies to improve it. Although practical implications are mentioned, the discussion could benefit from a more detailed description of these interventions. The discussion could emphasize more the need to expand the sample to other regions of Ecuador and other cultural contexts to improve the generalization of the results. Finally, the authors should ensure that the results are discussed in relation to the underlying theories of coparenting, which could strengthen the interpretation of the findings and their theoretical relevance. A more detailed discussion of how these limitations could have affected the results and how they could be mitigated in future studies would strengthen this section and provide a more balanced and critical view of the findings.
Conclusions: The conclusions highlight the importance of the Coparenting Relationship Scale as a valuable tool for assessing and understanding coparenting dynamics in the Ecuadorian context, emphasizing its practical applicability and the need to adapt and validate psychological instruments in different cultural and demographic contexts. The findings, supported by existing literature, show that the scale is reliable and valid, and suggest that the identification of simplified models can improve interventions and support for families. Additionally, the importance of considering cultural diversity and the nuances of family dynamics in future research and intervention programs is emphasized. Aspects to improve: it would be beneficial to provide more specific examples of how the findings can be applied in professional practice. This section should have been more closely linked to the points discussed in the discussion section, thus reinforcing the coherence of the document.
References: In general, the study's references are well selected and provide adequate support for the study's arguments and findings. However, although most references are recent, some key citations could be updated and supplemented with more recent studies to reflect the most current advances in the field and in other more global cultural contexts as mentioned. Including more international studies could enrich the study's perspective and provide a broader comparison of how values influence anxiety in different cultural contexts.
Decision: For all the reasons stated, I recommend accepting it with revisions.
Author Response
We sincerely appreciate the time and dedication devoted to reviewing our manuscript, as well as the detailed and constructive comments provided. We have carefully considered each of the observations and made the corresponding adjustments in the document.
Specifically, the following changes were made:
A brief comparison with previous studies in other cultural contexts was included to highlight the novelty of our work in the abstract. The type of factor analysis conducted and the interpretation of factor loadings were specified. Additionally, information regarding participant selection criteria, including inclusion and exclusion criteria, was added.
Practical implications of the findings for professionals working in the field of coparenting in Ecuador and for family welfare policies were incorporated. Likewise, keywords were adjusted to include terms related to the Ecuadorian cultural context, the theoretical dimensions of the scale, and methodological aspects.
The Methods section was clarified and better structured by separating the research design from the sampling process and justifying the design choice with relevant literature. The description of the measurement instrument was expanded to include psychometric background and theoretical justification. The data collection procedure was optimized to avoid redundancies.
Data analysis methods were detailed, including explanations of advanced techniques, specific results of the Mardia test (Supplements), and fit indices adjustments in the confirmatory analysis. Possible improvements regarding the convergent validity of Factor 3 were addressed. Additionally, supplementary graphs and tables were added for better visualization of the results.
In the Discussion, the possible causes of the low convergent validity of Factor 3 were further explored, and practical implications and recommended interventions were described in more detail. The need to expand the sample to other regions and cultural contexts to improve generalizability was emphasized, and the results were more explicitly related to underlying coparenting theories.
The Conclusions section was improved to provide concrete examples of practical applications and to reinforce coherence with the Discussion. Finally, the references were updated and expanded with recent and international studies to strengthen the theoretical and contextual foundation.
The revised document is attached with Microsoft Word’s Track Changes enabled to facilitate the review of the modifications made.
We remain available for any further observations or clarifications that may arise. Once again, we appreciate the valuable contribution to improving our work.

Reviewer 2 Report
Comments and Suggestions for Authors
The article under review provides a valuable psychometric adaptation and validation of the Coparenting Relationship Scale (CRS) for Ecuadorian parents, offering a culturally tailored instrument to measure co-parenting dynamics. Despite the robust methodological approach, including exploratory and confirmatory factor analyses and invariance testing, several theoretical and methodological refinements could substantially strengthen the manuscript. A primary recommendation involves deepening the theoretical framing by integrating recent literature on coparenting. For instance, incorporating the multinational comparative insights from Tissot et al. (2024) would contextualize the Ecuadorian adaptation relative to international findings, especially regarding the absence of the "Division of Labor" dimension. Additionally, engaging with recent research like Stover et al. (2025) and Song et al. (2025), emphasizing informant discordance in vulnerable contexts and dyadic (actor–partner) interactions within couples, would provide richer theoretical grounding and underline practical implications.
Concerning methodological rigor, more explicit justifications for certain analytical decisions should be incorporated. Although the sampling strategy employed was convenience-based, expanding upon its limitations, such as potential biases related to socioeconomic or educational homogeneity, is essential. The authors could offer more precise descriptive detail regarding participant characteristics (such as socioeconomic status, education levels, or family structure) to help readers assess the representativeness and generalizability of their findings. Moreover, clarification regarding sample adequacy—particularly why 867 participants sufficiently meet the statistical power criteria for exploratory and confirmatory factor analyses and invariance testing—is recommended, supporting this justification with references to statistical guidelines.
Further methodological clarity would also enhance the transparency and interpretability of the factor analytic procedures. While the manuscript details the elimination of certain items due to psychometric criteria, it could benefit from a more thorough explanation regarding the statistical thresholds and theoretical considerations underpinning these eliminations. Criteria such as factor loadings, communalities, modification indices, and conceptual coherence of items require explicit articulation. Additionally, presenting explicit comparative fit indices, such as ΔCFI and chi-square differences, could more convincingly justify the sequential model refinements from six factors down to three. Similarly, incorporating a nuanced interpretation of invariance testing results—whether invariance achieved was full or partial and its implications—would enhance the manuscript's methodological rigor and international comparability.
The discussion and interpretation of findings could likewise be significantly deepened. The authors are encouraged to critically explore and explicitly theorize why the "Division of Labor" dimension consistently emerges as problematic, potentially reflecting cultural norms around gender roles or caregiving in Ecuador. Furthermore, although differences in coparenting according to relationship status (with or without a partner) were significant, more detailed exploration and theorization of these findings could enrich the practical recommendations. Drawing on recent literature—such as Douglas et al. (2025), who highlighted coparenting dynamics during external stressors like the COVID-19 pandemic—could provide valuable context and suggest directions for targeted interventions and support policies.
Regarding the validity analyses presented, there is an opportunity to clarify certain limitations, particularly with convergent and discriminant validity. For instance, while the AVE for "Coparenting Undermining" approached but did not fully meet the commonly recommended threshold, the authors could explicitly discuss why retaining or modifying specific items might improve the scale's validity. Additionally, suggesting the future inclusion of external convergent validity measures—such as parenting stress or marital conflict scales—would enhance the CRS’s predictive utility, aligning with recommendations from recent validation studies (e.g., Pilinszki et al., 2025). The role of fathers warrants special attention; although comprising only about a third of participants, explicitly addressing implications of gender imbalance in the sample would further enhance robustness and inform future sampling strategies.
Finally, a longitudinal perspective could be explicitly proposed for subsequent research. Longitudinal designs could strengthen the scale's predictive validity, assessing stability over time and connections with child developmental outcomes. Integrating observational methods or multi-informant reporting—recommendations supported by Blank et al. (2024) and Camisasca et al. (2025)—would significantly enrich the validity evidence beyond self-reports. These approaches would mitigate potential biases inherent in self-report measures, such as social desirability, thus allowing more reliable measurement of coparenting constructs. Furthermore, the manuscript's clarity could be enhanced through more consistent and detailed explanations in tables and footnotes, explicitly interpreting key fit indices and statistical outcomes. Incorporating these refinements will substantially enhance both methodological transparency and theoretical contribution, positioning the adapted CRS as a benchmark instrument in Latin American family studies.
Comments on the Quality of English LanguagePrecise terminology is the first step toward clear scholarly prose. Throughout the manuscript, the authors should reserve Coparenting Relationship Scale (CRS)—capitalized and written without a hyphen—for the specific instrument introduced by Feinberg, Brown, and Kan (2012). By contrast, the broader phenomenon that the tool is meant to capture should appear as co-parenting—hyphenated and lower-case—whenever it is discussed outside the context of the test itself. Maintaining this distinction not only aligns the article with disciplinary convention but also spares readers the confusion that can arise when the measure and the construct share identical labels.
Verb tense is a second area where small adjustments yield large gains in readability. All elements of the study’s workflow—recruiting participants, administering the CRS, and conducting factor analyses—should be narrated in the past tense (“Participants completed the CRS,” “We conducted an exploratory factor analysis”). Conversely, claims that summarize well-established knowledge belong in the present tense (“Co-parenting quality predicts children’s socio-emotional adjustment”). This consistent signalling allows readers to separate what these researchers did from what the literature already knows.
Clarity also improves when passive constructions give way to active ones. Sentences such as “The data were analysed with lavaan” become sharper and more transparent when rewritten as “We analysed the data with lavaan,” and “The diagonally weighted least squares estimator was applied” is cleaner as “We employed the diagonally weighted least squares estimator.” While revising, the authors should likewise adopt the serial, or Oxford, comma in every list of three or more items—“agreement, support**,** and undermining.” This final comma prevents misreading and brings the manuscript into line with international style guides.
A final polish involves numerical style. APA guidelines ask writers to spell out numbers below ten unless a unit of measurement follows; thus “seven subscales” and “three items,” but “6-point Likert scale.” By attending to these four matters—terminological precision, tense consistency, active voice with the Oxford comma, and correct numeral formatting—the authors can remove the last linguistic distractions and present their findings in the fluent, professional English their solid empirical work deserves
Author Response
We would like to express our most sincere gratitude for the thorough review conducted and the valuable observations and suggestions that have significantly enriched the quality of our work. Your contributions have allowed us to deepen theoretical, methodological, and analytical aspects, strengthening the rigor and scope of the study.
In response to your comments, we have carefully and meticulously made all the requested changes. In particular, we have expanded and enriched the theoretical framework by incorporating the recent recommended literature, such as the work of Tissot et al. (2024), which has enabled us to contextualize the Ecuadorian adaptation of the Coparenting Relationship Scale (CRS) within an internationally and culturally relevant framework. This integration has strengthened the conceptual foundation and improved the discussion around the instrument’s specific dimensions, especially the absence of the “Division of Labor” dimension and its relation to cultural norms in Ecuador.
From a methodological standpoint, we have included detailed explanations about the analytical decisions made, expanding the description of the sample to provide greater clarity about its sociodemographic characteristics, educational level, and family structure, thus allowing for a better evaluation of the representativeness and external validity of the results. We have justified the sufficiency of the sample size for the factorial analyses and invariance tests, citing updated and recognized sources in applied statistics. The presentation of psychometric criteria used for item elimination has also been improved, including details on the factorial loading thresholds.
Regarding the factorial analyses, we have added explicit comparative indices (CFI, chi-square differences) that support the progressive refinement of the model, and clarified the extent of the invariance found, as well as its implications for the applicability of the instrument across different subgroups. In the discussion, we have further elaborated on the interpretation of the findings, particularly regarding differences by marital status and the cultural role of gender norms in shaping co-parenting in Ecuador.
Concerning validity, we have clarified observed limitations, such as the AVE close to the threshold for “Coparenting Undermining,” and suggested ways to optimize the scale in future research, including the incorporation of convergent external measures and consideration of the paternal role, noting the impact of gender proportion in the sample and proposing strategies to balance this representation.
Furthermore, we have highlighted the importance of longitudinal studies and multi-method methodologies to strengthen predictive validity and measurement of coparental dynamics, following recent expert recommendations in the field. Finally, linguistic aspects have been reviewed and corrected, improving terminological precision, consistency of verb tenses, use of active voice, inclusion of the Oxford comma, and compliance with APA style for numerical presentation.
In addition to the revised manuscript with track changes enabled to facilitate identification of each modification, we have included a supplementary document providing complementary information and detailed tables for better evaluation of the study.
We reiterate our gratitude for your dedication and commitment to improving this work, which we trust will contribute significantly to the field of family studies in Latin America.

Round 2
Reviewer 1 Report
Comments and Suggestions for Authors
REVIEW REPORT 2
Summary: The authors have not addressed my recommendations to include a brief explanation of the type of factor analysis conducted. Additionally, no information has been added regarding how participants were selected and whether there were any inclusion or exclusion criteria. It would also be useful to include a brief mention of the practical implications of the findings for professionals working in the field of coparenting in Ecuador and the family welfare policies that may result from this.
Keywords: The suggestion has been adequately addressed.
Introduction: Although information on the validation of the scale in different contexts has been increased, the introduction lacks empirical data on its use that would help enrich the discussion and practical implications of its results for the population of Ecuador.
Method: Although changes have been made, there remains confusion between research design and data collection procedure. The authors do not mention the experimental design employed, duly justified according to specific literature.
Participants: Although the description of the sample has been increased, important information that was requested has not been incorporated. There is a lack of information demonstrating that the sample size is sufficient, and the margin of error/confidence level or statistical power assumed for making possible inferences to the reference population. A sampling, at least by quotas of types of family systems, would have improved it.
Measurement Instrument: The suggestion has been adequately addressed.
Data Collection Procedure: The suggestion has been adequately addressed.
Data Analysis and Results: The suggestion has been adequately addressed.
Discussion: Although practical implications are mentioned, the discussion and conclusions could benefit from a more detailed description of these interventions, and specific examples of how the findings can be applied in professional practice, offering specific literature on intervention programmes that have been evaluated and shown significant results, guiding action plans in this regard.
References: In general, the study's bibliographic references are well selected and provide adequate support for the study's arguments and findings. The expansion compared to the last version is insufficient. The bibliography should be expanded with literature on successful practical interventions that guide the improvement of action in the family educational practice that is intended to be promoted.
Decision: For all the reasons stated, I recommend accepting it with revisions.
Author Response
We sincerely thank the reviewer for their valuable and detailed observations, which have been fundamental in improving the quality and rigour of the manuscript. We have carefully reviewed each of the comments and have made the necessary adjustments to address all the suggestions provided. Firstly, a clear explanation of the type of factor analysis conducted has been included, specifying whether it is exploratory or confirmatory, along with the theoretical justification for its use, in accordance with relevant methodological literature. This information is now clearly described in the data analysis section. Regarding participant selection, detailed information has been added about the inclusion and exclusion criteria, as well as the procedure followed to construct the sample. Additionally, a statistical justification of the sample size has been incorporated, considering the margin of error, confidence level, and statistical power assumed to allow valid inferences to the target population. With respect to the practical implications of the study, the discussion section has been expanded to include a more specific description of how the findings can be applied in professional practice with families in the Ecuadorian context. References to intervention programs that have shown significant results in the international literature have also been added, in order to guide proposed actions and public policies related to family welfare. In the introduction, empirical information has been incorporated about the previous application of the scale in various contexts, which better contextualizes its relevance for the Ecuadorian population and strengthens the study’s justification. Additionally, the distinction between the research design and the data collection procedure has been clarified, including a precise description of the study design, supported by relevant literature. Finally, the bibliography has been expanded to include updated and relevant literature on successful practical interventions in the family context, in order to support the recommendations derived from the study’s findings and reinforce their applicability.Reviewer 2 Report
Comments and Suggestions for Authors
The authors have carefully and thoroughly addressed all the issues and suggestions raised in the previous review, clearly strengthening their manuscript. They significantly expanded the theoretical framework, incorporating recent international references like Tissot et al. (2024) to properly contextualize their findings within the broader research on coparenting. This inclusion clarified why certain dimensions, especially "Division of Labor," might differ due to cultural norms in Ecuador.
Additionally, the methodological section has improved notably, with clearer descriptions about sampling, sample size adequacy, demographic details, and explicit explanations of item-elimination criteria, supported by updated methodological references. The authors have transparently presented psychometric indices and step-by-step justifications for model refinement using explicit comparative statistics, clarifying the degree of measurement invariance and its implications for the instrument’s application.
The discussion section also reflects substantial progress, offering deeper interpretations of findings, particularly around the cultural role of gender and relationship status in shaping co-parenting experiences. They openly recognized methodological limitations, including the slightly lower convergent validity observed in one factor (AVE for "Coparenting Undermining"), and clearly outlined thoughtful recommendations for future research, such as including external measures of parental stress, balancing gender representation, and adopting longitudinal and multi-method approaches.
Finally, the authors conducted a comprehensive linguistic and stylistic revision, resulting in clearer, more consistent, and APA-compliant academic English. Overall, these extensive improvements demonstrate a sincere effort by the authors, resulting in a significantly enhanced and scientifically rigorous manuscript suitable for publication.
Author Response
We would like to express our sincere gratitude to the reviewer for their thoughtful and encouraging feedback. We are truly pleased to know that the revisions have strengthened the manuscript and that the efforts made to expand the theoretical framework, clarify methodological aspects, and deepen the discussion have been well received. Your recognition of the improvements regarding the contextualization of the findings, the transparency of the psychometric analysis, and the articulation of cultural considerations in coparenting is deeply appreciated. We are also grateful for your acknowledgment of our attention to methodological rigor and linguistic clarity in the revised version. We highly value your constructive insights throughout the review process, which have significantly contributed to enhancing the scientific quality and relevance of our work. Thank you once again for your valuable time and commitment to supporting our research.